

# Effects of fertilizations on soil bacteria and fungi communities in a degraded arid steppe revealed by high through-put sequencing

Luhua Yao[1], Dangjun Wang[1], Lin Kang[1], Dengke Wang[1], Yong Zhang[2], Xiangyang Hou[2] and Yanjun Guo[1]

[1] College of Agronomy and Biotechnology, Southwest University, Chongqing, China
[2] Institute of Grassland Research, Chinese Academy of Agricultural Sciences, Hohhot, China

## ABSTRACT

**Background**. Fertilization as one of the measures in restoring degraded soil qualities has been introduced on arid steppes in recent decades. However, the fertilization use efficiency on arid steppes varies greatly between steppe types and years, enhancing uncertainties and risks in introducing fertilizations on such natural system to restore degraded steppes.

**Methods**. The experiment was a completely randomized design with five fertilization treatments, 0 (Control), 60 kg P ha$^{-1}$ (P), 100 kg N ha$^{-1}$ (N), 100 kg N ha$^{-1}$ plus 60 kg P ha$^{-1}$ (NP), and 4,000 kg sheep manure ha$^{-1}$ (M, equaling 16.4 kg P ha$^{-1}$ and 81.2 kg N ha$^{-1}$). Soils were sampled from a degraded arid steppe which was consecutively applied with organic and inorganic fertilizers for three years. We analyzed the diversity and abundance of soil bacteria and fungi using high-throughput sequencing technique, measured the aboveground biomass, the soil chemical properties (organic carbon, available and total phosphorus, available and total nitrogen, and pH), and the microbial biomass nitrogen and microbial biomass carbon.

**Results**. In total 3,927 OTU (operational taxonomic units) for bacteria and 453 OTU for fungi were identified from the tested soils. The Ace and Chao of bacteria were all larger than 2,400, which were almost 10 times of those of fungi. Fertilizations had no significant influence on the richness and diversity of the bacteria and fungi. However, the abundance of individual bacterial or fungi phylum or species was sensitive to fertilizations. Fertilization, particularly the phosphorus fertilizer, influenced more on the abundance of the AMF species and colonization. Among the soil properties, soil pH was one of the most important soil properties influencing the abundance of soil bacteria and fungi.

**Discussion**. Positive relationships between the abundance of bacteria and fungi and the soil chemical properties suggested that soil bacteria and fungi communities in degraded steppes could be altered by improving the soil chemical properties through fertilizations. However, it is still not clear whether the alteration of the soil microbe community is detrimental or beneficial to the degraded arid steppes.

Corresponding author
Yanjun Guo, qhgyj@swu.edu.cn, qh-gyj@126.com

# INTRODUCTION

Steppe is the largest land ecosystem in China, which plays important roles in providing life necessities, ecological services and recreations for human beings (*Du et al., 2014*; *Ren, Lu & Fu, 2016*). However, long-term drought, frequent human activities and overgrazing have caused severe steppe degradation such as decreases of soil qualities and productivities (*Zhao et al., 2009*; *Bai et al., 2012*). Fertilization as one of the measures in restoring degraded soil qualities has been introduced on steppes in recent decades, mainly concentrating on its efficiency in improving soil nutrient conditions (*Gong et al., 2011*), increasing aboveground biomass and vegetation coverage (*Zhou et al., 2016*; *Yang, Ruijven & Du, 2011*), and enriching ecological biodiversity (*Symstad & Jonas, 2011*). However, the fertilization use efficiencies in steppes varied greatly between steppe types and years (*Su, Li & Yang, 2014*; *Zhou et al., 2016*; *Yang, Ruijven & Du, 2011*). This increased the uncertainties in restoring the degraded steppes through fertilization and the risk in introducing more fertilization into such natural system.

Fertilizations influence both the aboveground biomass and the belowground microbe biomass. Soil microbial communities play critical roles in ecosystem function and regulate key processes such as carbon and nitrogen cycles (*Balser & Firestone, 2005*; *Bragazza et al., 2015*). In a Tibetan alpine meadow, nitrogen (N) and phosphorus (P) additions reduced the biomass of the whole microbial community, gram negative bacteria and fungi (*Guo et al., 2017*). In a subtropical forest in East China, soil microbial biomass was enhanced by mixed N fertilization (*Guo et al., 2011*). The different responses of the soil microbes to fertilizations, on one hand, are attributed to the differences in the contents of soil total carbon (C), N and P (*Yu et al., 2013*), and the differences of soil water conditions and plant species (*Huang et al., 2018*). On the other hand, the soil microbe communities in different ecosystems are different, and thus their responses to similar fertilizations might also be different.

During the long-term evolution process, the soil, plants and microbes have co-evolved, forming relatively stable relationships under certain ecosystem (*terHorst, Lennon & Lau, 2014*). In this relationship, genetic variation in plant phenotypes can change soil processes and biotic communities, whereas soil gradients and microbial communities can influence the expression and evolution of plant phenotypes (*Van Nuland et al., 2016*). When exterior N and P are input into local ecosystem, the given balance between soils, plants and microbes will be destroyed, and a new balance will be formed. For example, in a 50-year-old fertilization experiment, the composition of soil nitrate-reducing, denitrifying and total bacterial communities co-varied with primary production and both were strongly linked to soil properties (*Hallin et al., 2009*). In paddy soils, *Bai et al. (2015)* reported that anammox was more common and active in alkaline soils than in acidic soils, and the anammox activities were likely to be regulated by soil chemical properties such as pH, salinity and redox potential.

Soil microbes are mainly consisted of bacteria, fungi and Archaea. They are involved in catalyzing the transitions between compounds of N, P and C, benefiting the plant uptake and microbial propagation (*Van Nuland et al., 2016*; *Zhang et al., 2017*). In a paddy soil, the

application of organic fertilizer increased the abundance of the copiotrophic bacteria such as Proteobacteria phylum, and the relative abundance of Agaricomycetes and Orbiliomycetes classes of fungi (*Wang et al., 2017*). In an arable soil, arbuscular mycorrhiza fungi (AMF) diversity and richness significantly decreased under long-term P fertilizations (*Lin et al., 2012*). In a temperate steppe in Inner Mongolia, N fertilization changed ammonia-oxidizing bacteria (AOB) community composition and increased AOB abundance in both May and August, but significantly decreased ammonia-oxidizing archaea (AOA) abundance in May (*Chen et al., 2014*). In a temperate desert, nitrogen fertilization increased the relative abundance of Ascomycota phylum and decreased Actinobacteria phylum in spring (*Huang et al., 2018*). Such changes in soil microbe communities induced by environmental changes might have detrimental effects on the relationship between soil microbes and the plants, and thus negatively influence the productivity. For example, the changes in the fungal to bacterial (F:B) ratio in the soil microbial biomass can lead to increases in total microbial community C:N and declines in biomass turnover rates, both of which can reduce microbial N uptake (*Waring, Averill & Hawkes, 2013*). The decrease of AMF diversity and abundance after fertilization will limit plant nutrient uptake under low soil phosphorus and drought conditions (*Wu et al., 2011a*).

The marker gene amplification metagenomics represent an important way to acquire information on the microbial communities present in complex environments like soil (*Lombard et al., 2011*), which allow to identify soil microbes that are unable to be cultured in medium. Marker gene amplification metagenomics, typically using the 16S and 18S ribosomal RNA, have been applied in studying the uncultured microbial populations in various environments (*Handelsman, 2009*; *Oulas et al., 2015*). Therefore, in the current study, using high-throughput sequencing technique, we analyzed the soil microbe communities including bacteria and fungi in soils of a degraded steppe which has been fertilized for consecutively three years. The responses of the microbial biomass nitrogen (MBN) and microbial biomass carbon (MBC) as well as the aboveground biomass and the soil chemical properties to fertilizations were also measured. The main objective of the current study was to clarify how the soil microbe communities respond to three years' fertilization on a degraded arid steppe and their relationship with soil chemical properties and aboveground biomass.

## METHODS

### Study site

The experimental site is located in Arigalangtu (43°50′23″N;116°09′57″E), 15 km away from Xilinhaote, Inner Mongolia, China. Average annual precipitation is 350 mm and average annual temperature is 1.7 °C during the last 50 years, whereas the precipitation in 2014, 2015 and 2016 were 255 mm, 412 mm and 309 mm, respectively. The vegetation is a typical steppe, located in an open high plain with altitude reaching approximately 1,290 m. Based on plant species investigation, *Stipa krylovii* is the dominant species, dry weight of which accounts for ca. 68%, followed by *Allium bidentatum* and *Leymus chinensis*, accounting for ca. 18%, with the remaining consisting of *Convolvulus ammannii, Cleistogenes squarrosa,*

*Salsola collina*, *Agropyron michnoi*, and *Carex korshinskyi, et al*. The soil is mainly light Kastanozems (FAO soil classification). Soil texture averages 21% clay, 30% silt and 49% sand. This area has been continuously grazed by sheep and cattle for at least 20 years before 2014, showing severe deleterious symptoms, mainly low plant productivity and low plant coverage, typical in the most degraded steppes in Inner Mongolia (*Pei, Fu & Wan, 2008*; *Schönbach et al., 2011*).

In June 2014, about 50 km$^2$ of this degraded steppe were fenced and excluded from grazing. Soils (0–20 cm) were sampled and analyzed for basic chemical properties. The concentrations of alkali dispersed N (AN), available phosphorus (AP, Olsen-P) and available potassium (AK) were 69.79, 3.32 and 133.07 mg kg$^{-1}$, respectively. The concentrations of total nitrogen (TN), total phosphorus (TP), total potassium (TK), and organic carbon (OC) were 1.16, 0.34, 16.21, and 17.75 g kg$^{-1}$. The soil pH was 7.75 (soil:water = 1:5).

## Experimental design

The experiment was a completely randomized design with five fertilization treatments, 0 (Control), 60 kg P ha$^{-1}$ (P), 100 kg N ha$^{-1}$ (N), 100 kg N ha$^{-1}$ plus 60 kg P ha$^{-1}$ (NP), and 4,000 kg sheep manure ha$^{-1}$ (M, equaling 16.4 kg P ha$^{-1}$ and 81.2 kg N ha$^{-1}$). The inorganic phosphorus fertilizer was superphosphate (14% P); the nitrogen fertilizer was urea (46% N); the sheep manure was collected from farmer's feedlot and composted for 3 months (April to June), air dried, and sieved (<2 mm) before being applied to the field. Each treatment replicated three times with random complete block design. Each plot size reached 400 m$^2$ (20 m $\times$ 20 m). Fertilizers were applied on steppe surface in rainy day for three times with once in each year, July 2014, June 2015 and June 2016. The steppes were cut for hay once a year at the end of growing season in late August and early September each year.

## Aboveground biomass and root samples

Aboveground biomass was measured in each month during the growing season, June, July and August of 2016. In each of the sampling plot, plants were cut at the ground level from three quadrats (1 m $\times$ 1 m), dried at 75 °C for 48 h, and weighed. Meanwhile, fine roots from three plant species, *Cleistogenes squarrosa*, *Stipa krylovii* and *Leymus chinensis*, were sampled, cleaned, and stored in FAA (5 mL formalin (38% formaldehyde ) + 5 mL acetic acid + 90 mL alcohol (70%)) solution.

## Collection of soil samples

After the aboveground biomass were harvested, soil samples were collected in June (before fertilization), July (one month after fertilization) and August (two month after fertilization) 2016. In total, nine soil cores (6 cm in diameter) from three quadrats in each plot were collected at 0–10 cm and 10–20 cm soil layers in each month, bulked into one composite soil sample for each plot from each soil layer. The soils were further divided into two parts, part one was air-dried, sieved to 2 mm for soil chemical property analysis; part two was stored in 4 °C for MBC and MBN analysis. For soil samples in July 2016, ca. 100 g of fresh soil per sample was stored at −80 °C for DNA extraction. July was the growing month that most plant species were in their vigorous growth stage.

## Soil chemical analysis

Organic carbon (OC) was determined by oxidation with potassium dichromate in a concentrated sulfuric acid medium and the excess dichromate was measured using Mohr's salt ($K_2Cr_2O_7$–$H_2SO_4$) (*Yeomans & Bremner, 1988*). Dried soils (1.000 g) were digested in 5 mL $H_2SO_4$ and then determined for soil total nitrogen (TN) by Kjeldahl method (*Bao, 2005*). Total phosphorus (TP) was measured using NaOH fusion and Mo-Sb colorimetric procedure (*Bao, 2005*). Total potassium was measured using NaOH fusion and flame photometry (*Bao, 2005*). Dried soils (2.500 g) were mixed and shaken in 50 mL 0.5 mol $L^{-1}$ $NaHCO_3$ for 30 min, and then the supernatant was analyzed for available phosphorus (AP) using Mo-Sb colorimetric procedure (*Bao, 2005*). Dried soils (2.000 g) were digested in 10 mL 1 mol $L^{-1}$ NaOH solution for 24 h and titrated with 0.01 mol $L^{-1}$ 1/2 $H_2SO_4$ for alkali dispelled nitrogen (AN) (*Bao, 2005*). Dried soils (5.000 g) were digested in 50 mL 1 mol L-1 $NH_4Ac$ solution for 15 min, and then the supernatant was analyzed for available potassium (AK) using flame photometer method (*Bao, 2005*). Soil pH value was determined in a soil:water solution (1:5) using a pH meter (*Bao, 2005*).

The soil microbial biomass carbon (MBC) and microbial biomass nitrogen (MBN) was measured by fumigation-extraction procedure (*Brookes et al., 1985*). Ten grams of fresh soil samples fumigated with chloroform and non-fumigated were extracted with 50 ml of 0.5 mol $L^{-1}$ $K_2SO_4$ separately. Total carbon and nitrogen in the subsequent filtrates were measured by the methods used for soil samples as mentioned above.

$$MBC = (A - B) \div 0.38$$

*A*: extractable organic C in fumigated.
*B*: extractable organic C non-fumigated soil.
0.38: The conversion factor for microbial biomass carbon.

$$MBN = (C - D) \div 0.45$$

*C*: extractable organic N in fumigated.
*D*: extractable organic N non-fumigated soil.
0.45: The conversion factor for microbial biomass nitrogen.

## Assessment of AMF colonization

Approximately 2 g of fine roots were cut into 1 cm pieces, cleared in 2% KOH (w/v) at 90 °C for 60 min and rinsed three times in water on a fine sieve. The root samples were acidified in 2% HCl (v/v) for 30 min and then stained in 0.05% (w/v) trypan blue in lactoglycerol for 30 min at 90 °C. Root segments of each species were selected randomly from the stained samples. Four replicates of 15 roots per slide were assessed for the presence or absence of AMF structures (arbuscules, vesicles and thick hyphae) using a stereomicroscope. AMF colonization was distinguished from non-mycorrhizal fungi as described by *Callaway et al. (2003)*. Colonization was expressed as mycorrhizal frequency (F%), mycorrhizal intensity (M%) and arbuscular density (A%) according to the method of *Trouvelot et al. (1986)*.

## DNA extraction and PCR amplification

Microbial DNA was extracted from 30 samples using the E.Z.N.A.® Kit (Omega Bio-tek, Norcross, GA, US) according to manufacturer's protocols. The V3-V4 region of the bacteria 16S ribosomal RNA gene were amplified by PCR (95 °C for 3 min , followed by 27/35 cycles at 95 °C for 30 s, 55 °C for 30 s, and 72 °C for 45 s and a final extension at 72 °C for 10 min, 10 °C until halted by user) using primers 338F (5′-ACTCCTACGGGAGGCAGCA-3′) and 806R (5′-ATGCAGGGACTA CHVGGGTWT CTAAT-3′). The fungal sequence of 18S rRNA genes was amplified using primers SSU0817F (5′-TTAGCATGGAATAATRRAATAGGA-3′) and SSU1196R (5′-TCTGGACCT GGTGAGTTTCC-3′). PCR reactions were performed in triplicate 20 µL mixture containing 4 µL of 5 × FastPfu Buffer, 2 µL of 2.5 mM dNTPs, 0.8 µL of each primer (5 µM), 0.4 µL of FastPfu Polymerase, and 10 ng of template DNA. Bacterial and fungal PCR products were pooled separately to be sequenced in their runs.

## Illumina MiSeq sequencing

Amplicons were extracted from 2% agarose gels and purified using the AxyPrep DNA Gel Extraction Kit (Axygen Biosciences, Union City, CA, US) according to the manufacturer's instructions and quantified using QuantiFluor™-ST (Promega, Madison, WI, USA). Purified amplicons were pooled in equimolar and paired-end sequenced (2 × 300) on an Illumina MiSeq platform (Illumina, San Diego, CA, USA) according to the standard protocols. All the sequences of bacteria and fungi were deposited into the NCBI Sequence Read Archive database (SRP126759).

## Statistical analysis

Raw fastq files were demultiplexed, quality-filtered by Trimmomatic and merged by FLASH with the following criteria: (i) the reads were truncated at any site receiving an average quality score <20 over a 50 bp sliding window. (ii) Primers were exactly matched allowing 2 nucleotide mismatching, and reads containing ambiguous bases were removed. (iii) Sequences whose overlap longer than 10 bp were merged according to their overlap sequence. Operational Taxonomic Units (OTUs) were clustered with 97% similarity cutoff using UPARSE (version 7.1 http://drive5.com/uparse/) and chimeric sequences were identified and removed using UCHIME. The OTU representative sequences were analyzed by RDP Classifier (http://rdp.cme.msu.edu/) against the silva (SSU123)16S rRNA database using confidence threshold of 70% (*Amato et al., 2013*), while the representative sequences of the fungi were analyzed using UNITE database (*Abarenkov et al., 2010*).

The alpha and beta diversity analyses were performed using the high quality sequences. To assess the diversity of the samples, we used the observed richness estimator Sobs, the coverage estimator Ace, the richness estimator Chao1, the Good's coverage and the Shannon and Simpson diversity indices. To determine the effects of fertilization regimes on bacterial and fungal community compositions, permutational multivariate analysis was performed using the 'vegan' package in R software (*Team RC, 2014*). The correlation analysis of the relative abundance of the bacterial and fungal taxa was held at a phylum level (abundance of the top 50 species) using the R package.

Statistical analyses were performed with SPSS 17.0 (SPSS, Inc., Chicago, IL, USA). Results were expressed as mean ± standard error (SE). Nonparametric variables were mathematically transformed to improve symmetry. The correlation between relative abundance of certain OTU/genus/phylum and environmental factors was performed by Spearman's correlation analyses. $P < 0.05$ was considered to be statistically significant according to least significant difference test. The Glomeromycota sequences were aligned and tested using the best-fit model with MEGA 4.0 (*Tamura et al., 2007*). For the phylogenetic analysis, the maximum likelihood method (*Nei & Kumar, 2000*) was used to construct the tree with MEGA 4.0 using DNA sequences selected according to their sequence similarity to the reference data in GenBank.

# RESULTS

## Soil chemical properties and aboveground biomass

The contents of soil OC and TN showed no significant difference between fertilization treatments at 0–10 cm or 10–20 cm and in three sampling months (Table 1). The ratios of C/N ranged from 6.61 to 13.84 and their responses to fertilization differed between sampling months and soil layers. At 0–10 cm soil layer, C/N ratio in non-fertilized plots was higher than those in fertilized plot in June, lower than those in P and N fertilized plots in July, and unchanged in August. The contents of AN changed insignificantly after fertilization and varied slightly between months, with the AN contents at 0–10 cm soil layer higher than those at 10–20 cm soil layer. The contents of AP were lower than 5 mg kg$^{-1}$ in non-fertilized plots in all sampling months, which increased after phosphorus or manure fertilization. The AP contents at 0–10 cm soil layer were higher than those at 10–20 cm soil layer. The soil pH level ranged from 7.37 to 7.85 at 0–10 cm soil layer and from 7.71 to 8.19 at 10–20 cm soil layer. Overall, though fertilizations had a trend in increasing the aboveground biomass in all sampling months, no significant difference could be observed between fertilized and non-fertilized plots (Table 2).

## Soil microbial biomass carbon and nitrogen

The influences of fertilizations on MBC and MBN varied between treatments and soil layers (Fig. 1). Overall, the MBC and MBN at 0–10 cm soil layer were higher than those at 10–20 cm soil layer. Phosphorus, nitrogen and manure fertilizations increased the MBC and MBN contents in all sampling months, excepting for insignificant changes of MBC in manure fertilized plot in July at both soil layers and insignificant changes of MBN in all fertilizations in July at 10–20 cm soil layer excepting NP treatment. In June, the MBC in N, NP and manure fertilized plots showed no significant differences, which were significantly lower than those in P fertilized plots at both soil layers. In July, the MBC in manure fertilized plots was significantly lower than those in P, N and NP fertilized plots at both soil layers. In August, the MBC in N and P fertilized plots was the highest and that in manure plots was the lowest at 0–10 cm soil layer, whereas the MBC in P fertilized plot was significantly higher than those in other treatments at 10–20 cm soil layer. The MBN was the highest in manure fertilized plots at both soil layers in June, showed no difference between treatments in July excepting for higher MBN in NP plot at 10–20 cm soil layer, and was the highest

**Table 1  Effects of fertilizations on the soil chemical properties.**

| De | Tr | OC g kg$^{-1}$ | TN g kg$^{-1}$ | C/N | AN mg kg$^{-1}$ | AP mg kg$^{-1}$ | pH |
|---|---|---|---|---|---|---|---|
| **June 2016** | | | | | | | |
|  | CK | 10.09 ± 0.23a | 0.87 ± 0.02a | 11.57 ± 0.05a | 98.70 ± 1.21a | 3.58 ± 1.31c | 7.83 ± 0.11a |
|  | P | 9.69 ± 0.23a | 0.98 ± 0.03a | 9.92 ± 0.36b | 100.80 ± 5.28a | 14.43 ± 0.51a | 7.75 ± 0.09a |
| 0–10 cm | N | 10.85 ± 0.68a | 1.07 ± 0.02a | 10.13 ± 0.57b | 103.60 ± 2.80a | 6.33 ± 2.35bc | 7.58 ± 0.16a |
|  | NP | 9.96 ± 0.13a | 1.16 ± 0.01a | 8.59 ± 0.13c | 107.10 ± 3.21a | 13.91 ± 0.55a | 7.75 ± 0.09a |
|  | M | 10.80 ± 0.29a | 1.12 ± 0.02a | 9.67 ± 0.17b | 101.50 ± 12.38a | 10.48 ± 3.27ab | 7.82 ± 0.06a |
|  | CK | 8.34 ± 0.59a | 0.88 ± 0.06a | 9.56 ± 0.56ab | 79.80 ± 4.85a | 1.83 ± 0.80b | 7.99 ± 0.09a |
|  | P | 7.77 ± 0.29a | 0.91 ± 0.05a | 8.53 ± 0.17bc | 83.13 ± 3.20a | 6.25 ± 1.09a | 7.71 ± 0.12a |
| 10–20 cm | N | 8.38 ± 0.09a | 0.94 ± 0.03a | 8.97 ± 0.36bc | 72.80 ± 1.40a | 4.17 ± 0.36ab | 7.82 ± 0.09a |
|  | NP | 7.67 ± 0.01a | 0.92 ± 0.01a | 8.33 ± 0.11c | 86.63 ± 6.26a | 2.83 ± 0.36ab | 8.02 ± 0.12a |
|  | M | 9.52 ± 0.12a | 0.91 ± 0.02a | 10.45 ± 0.38a | 72.80 ± 1.40a | 3.67 ± 2.10ab | 8.04 ± 0.08a |
| **July 2016** | | | | | | | |
|  | CK | 10.51 ± 0.40a | 1.05 ± 0.02a | 10.00 ± 0.47c | 88.90 ± 9.42a | 2.92 ± 0.10c | 7.85 ± 0.19a |
|  | P | 12.13 ± 0.47a | 0.88 ± 0.001a | 13.84 ± 0.55a | 97.30 ± 3.90a | 19.44 ± 1.83a | 7.75 ± 0.14a |
| 0–10 cm | N | 12.44 ± 0.18a | 1.03 ± 0.05a | 12.15 ± 0.64b | 98.70 ± 9.15a | 3.79 ± 0.83c | 7.37 ± 0.15a |
|  | NP | 12.13 ± 0.83a | 1.18 ± 0.04a | 10.28 ± 0.61c | 109.90 ± 5.47a | 11.25 ± 2.66b | 7.59 ± 0.16a |
|  | M | 10.83 ± 0.18a | 1.19 ± 0.05a | 9.12 ± 0.21c | 93.10 ± 8.94a | 4.23 ± 1.51c | 7.63 ± 0.15a |
|  | CK | 9.46 ± 0.24a | 0.88 ± 0.02a | 10.71 ± 0.12a | 72.10 ± 2.80a | 2.52 ± 0.12b | 8.05 ± 0.12a |
|  | P | 9.60 ± 0.14a | 0.85 ± 0.03a | 11.32 ± 0.47a | 74.90 ± 9.10a | 4.16 ± 0.38a | 8.19 ± 0.16a |
| 10–20 cm | N | 9.77 ± 0.65a | 0.91 ± 0.01a | 10.74 ± 0.78a | 77.70 ± 9.92a | 2.15 ± 0.34b | 7.86 ± 0.03a |
|  | NP | 8.53 ± 0.42a | 1.00 ± 0.03a | 8.55 ± 0.29b | 89.60 ± 6.22a | 2.75 ± 0.66b | 8.04 ± 0.09a |
|  | M | 10.38 ± 0.65a | 0.99 ± 0.06a | 10.51 ± 0.31a | 90.3 ± 10.57a | 1.98 ± 0.19b | 8.00 ± 0.07a |
| **August 2016** | | | | | | | |
|  | CK | 9.25 ± 0.05a | 1.26 ± 0.03a | 7.37 ± 0.18a | 92.17 ± 6.27a | 3.38 ± 0.57c | 7.78 ± 0.12a |
|  | P | 11.32 ± 0.08a | 1.23 ± 0.01a | 9.23 ± 0.14a | 94.73 ± 3.06a | 15.17 ± 3.47a | 7.64 ± 0.10a |
| 0–10 cm | N | 10.29 ± 0.87a | 1.42 ± 0.12a | 7.28 ± 0.41a | 106.6 ± 15.16a | 3.22 ± 0.50c | 7.45 ± 0.12b |
|  | NP | 9.00 ± 0.44a | 1.36 ± 0.01a | 6.61 ± 0.35a | 107.10 ± 8.17a | 11.85 ± 2.49ab | 7.44 ± 0.06b |
|  | M | 10.15 ± 0.00a | 1.23 ± 0.04a | 8.29 ± 0.24a | 92.26 ± 2.03a | 8.17 ± 2.23bc | 7.49 ± 0.03a |
|  | CK | 9.50 ± 0.16a | 1.09 ± 0.003a | 8.75 ± 0.12b | 80.73 ± 3.93a | 2.31 ± 0.30c | 8.07 ± 0.15a |
|  | P | 10.66 ± 0.21a | 1.02 ± 0.01a | 10.43 ± 0.15b | 77.93 ± 2.63a | 5.32 ± 1.64ab | 8.00 ± 0.08a |
| 10–20 cm | N | 8.78 ± 0.32a | 1.15 ± 0.07a | 6.88 ± 0.21b | 74.20 ± 10.01a | 2.28 ± 0.27c | 8.04 ± 0.28a |
|  | NP | 7.91 ± 0.13a | 1.12 ± 0.03a | 7.06 ± 0.27b | 72.80 ± 1.85a | 7.35 ± 1.02a | 7.92 ± 0.03a |
|  | M | 9.78 ± 0.02a | 1.00 ± 0.05a | 9.82 ± 0.50a | 78.26 ± 7.40a | 3.62 ± 0.18bc | 7.76 ± 0.01a |

**Notes.**

Different lowercase letters after the data within each soil layer represented significance at $P < 0.05$ (l.s.d).

De, depth; Tr, treatment; CK, Control; P, 60 kg P ha$^{-1}$; N, 100 kg N ha$^{-1}$; NP, 60 kg P ha$^{-1}$ plus 100 kg N ha$^{-1}$; M, 4,000 kg sheep manure ha$^{-1}$. OC, organic C; TN, total N; AN, alkali dispelled N; AP, available P. Values were the means of three replicates ± SE.

in NP plot at both soil layers in August. The ratio between MBC and MBN was all larger than 2. At 0–10 cm soil layer, the ratio increased in fertilized plots excepting for manure fertilized plots in all sampling months and NP fertilized plot in August. At 10–20 cm soil layer, the ratio increased only in P plots in June, in P, N and NP plots in July, and in P and N plots in August.

**Table 2  Effects of fertilizations on total aboveground biomass (kg DM ha⁻¹).**

|  | June | July | August |
|---|---|---|---|
| CK | $702.9 \pm 7.42a$ | $863.4 \pm 106.9a$ | $897.4 \pm 21.1a$ |
| P | $705.3 \pm 53.69a$ | $1{,}087.7 \pm 108.9a$ | $1{,}095.0 \pm 23.8a$ |
| N | $692.0 \pm 65.0a$ | $1{,}062.5 \pm 94.4a$ | $937.0 \pm 21.8a$ |
| NP | $755.1 \pm 36.1a$ | $1{,}131.0 \pm 46.1a$ | $1{,}089.2 \pm 9.5a$ |
| M | $747.3 \pm 79.5a$ | $1{,}013.9 \pm 81.8a$ | $856.6 \pm 122.2a$ |

**Notes.**
Different lowercase letter after the data in each month represented significance at $P < 0.05$ level (l.s.d.).
CK, Control; P, 60 kg P ha⁻¹; N, 100 kg N ha⁻¹; NP, 60 kg P ha⁻¹ plus 100 kg N ha⁻¹; M, 4,000 kg sheep manure ha⁻¹.

## Biodiversity and abundance of bacterial and fungi

In this study, 16S rRNA genes were used as phylogenetic makers to determine changes in bacterial and fungi community structure. In total 1,120,431 bacterial sequences and 1,129,387 fungi sequences were obtained from 30 soil samples, which were further clustered into 3,927 OTU for bacteria and 453 OTU for fungi, according to 97% similarities. Based on these OTUs, the Richness estimator, Ace and Chao, and diversity index, Shannon and Simpson, were further analyzed (Table 3). Overall, the Ace and Chao of bacteria were all larger than 2,400, which were almost 10 times of those of fungi. No significant differences of richness estimators could be observed for bacteria and fungi at both soil layers. The Shannon index of bacteria ranged from 6.16 to 6.49 for bacteria, significantly higher than those for fungi which were all lower than 3.5. The Simpson index of fungi ranged from 5.93 to 9.05, significantly higher than those of bacteria which were all lower than 0.8. Bacteria had higher Shannon index and lower Simpson index at 0–10 cm soil layer than those at 10–20 cm soil layer, whereas no obvious difference could be observed for fungi.

Cluster analysis based on sequencing data of bacteria and fungi indicated that manure fertilization had the greatest influence on soil microbe communities (Fig. 2). The soil bacterial sequences in manure fertilized plots were separated with those in the other treatments at both soil layers. For fungi, sequences in phosphorus fertilized plot were separated with other treatments at 0–10 cm soil layer, while those in manure fertilized plots were separated with other treatments at 10–20 cm soil layer.

## Communities of soil bacteria and fungi

In the tested soils, the dominant bacteria species with abundance larger than 1% included Actinobacteria (42.21%), Proteobacteria (18.44%), Acidobacteria (15.71%), Chloroflexi (8.27%), Gemmatimonadetes (4.06%), Verrucomicrobia (3.18), Firmicutes (2.21%), Bacteroidetes (1.79%), and Nitrospirae (1.34%) (Fig. 3A, Fig. S1). At 0–10 cm soil layer, manure fertilization had a trend in reducing the abundance of Nitrospirae ($P < 0.05$) and increasing the abundance of Proteobacteria ($P < 0.05$) (Table S1). At 10–20 cm soil layer, the abundance of Chloroflexi in manure fertilized plot was significantly higher than those in non-fertilized, P, N and NP fertilized plots.

The dominant fungi species with abundance larger than 1% included Ascomycota (71.26%), Basidiomycota (13.50%), Glomeromycota (4.67%), Zygomycota (4.56%), Ciliophora (1.60%), Chytridiomycota (1.25%) (Fig. 3B, Fig. S2). At 0–10 cm soil layer, the

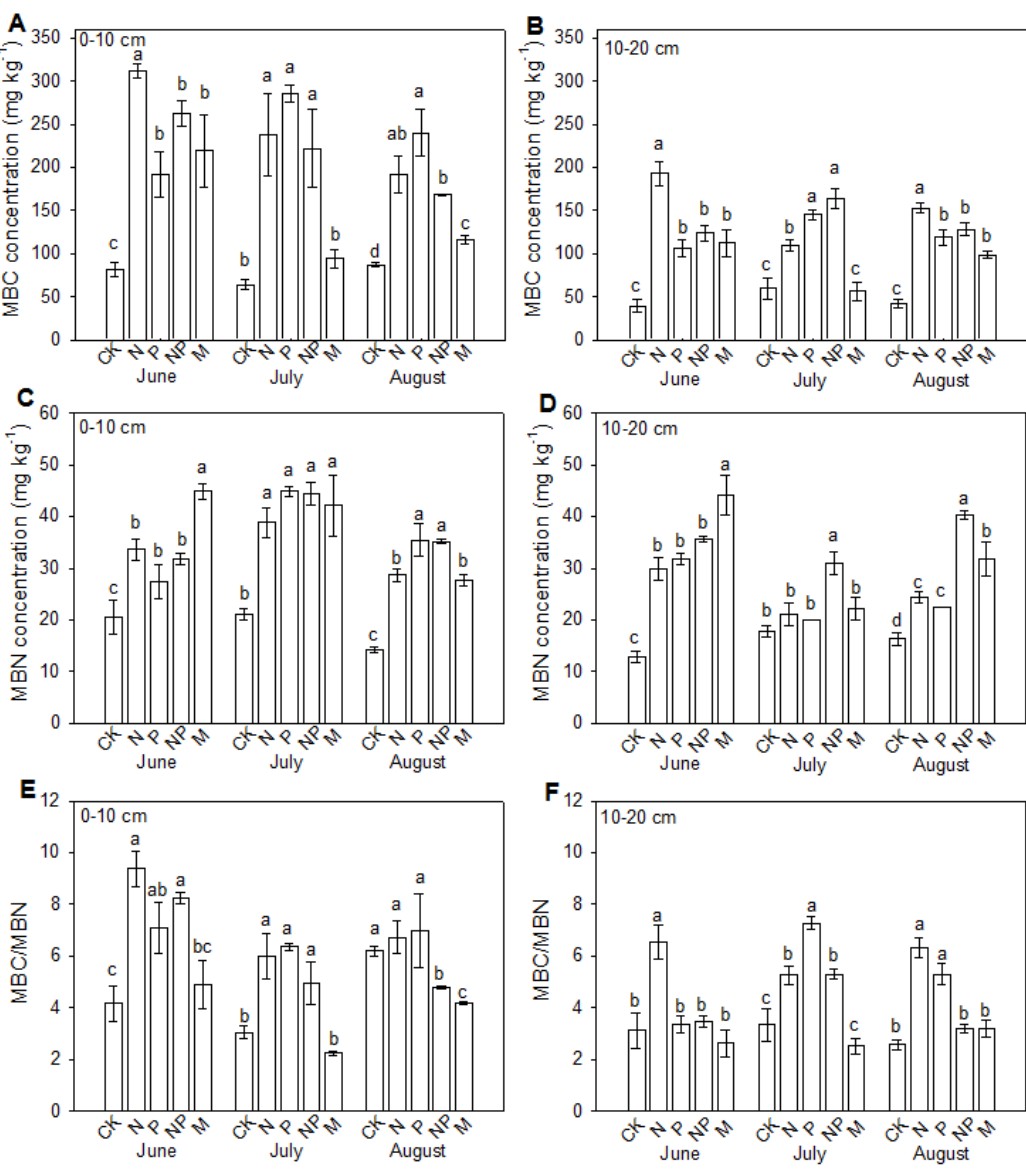

**Figure 1** **Effects of continuous fertilization on soil microbial biomass C and microbial biomass N at 0–10 cm (A, C, E) and 10–20 cm (B, D, F) soil layers.** CK, Control; P, 60 kg P ha$^{-1}$; N, 100 kg N ha$^{-1}$; NP, 60 kg P ha$^{-1}$ plus 100 kg N ha$^{-1}$; M, 4,000 kg sheep manure ha$^{-1}$. Data were the average of three replicates (±SE). Different lowercase letters above the data bar represented significance at $P < 0.05$ level (l.s.d).

abundance of Ciliophora was the highest in manure fertilized plot ($P < 0.05$) (Table S2). At 10–20 cm soil layer, the abundance of Chytridiomycota increased in P and NP treatments when compared with the non-fertilized plot, whereas unchanged in N and M treatments.

## AMF colonization and diversity
The fungi sequences were further blasted in NCBI and seven sequences matched with the published AMF sequences (Fig. 4). Cluster analysis separated these sequences into two

**Table 3  Diversity indices of soil bacterial and fungi communities in degraded arid steppes as influenced by fertilizations.**

| | | Treatment | Richness estimator | | Diversity index | | |
| --- | --- | --- | --- | --- | --- | --- | --- |
| | | | Ace | Chao | Shannon | Simpson (%) | Coverage (%) |
| 0–10 cm | Bacterial | CK | 2,752.09 ± 31.90a | 2,732.96 ± 44.32a | 6.49 ± 0.01a | 0.45 ± 0.01a | 96.97 |
| | | P | 2,695.87 ± 76.79a | 2,711.19 ± 82.01a | 6.42 ± 0.04a | 0.48 ± 0.06a | 96.96 |
| | | N | 2,654.69 ± 35.76a | 2,691.37 ± 33.05a | 6.36 ± 0.06a | 0.52 ± 0.09a | 97.02 |
| | | NP | 2,626.71 ± 81.73a | 2,635.74 ± 81.80a | 6.42 ± 0.11a | 0.44 ± 0.02a | 97.08 |
| | | M | 2,629.60 ± 29.36a | 2,612.58 ± 21.75a | 6.41 ± 0.05a | 0.46 ± 0.08a | 97.07 |
| | Fungi | CK | 182.89 ± 3.20a | 183.40 ± 2.76a | 3.49 ± 0.06a | 5.93 ± 0.81a | 99.91 |
| | | P | 164.53 ± 12.04a | 167.97 ± 15.55a | 3.36 ± 0.12a | 6.19 ± 0.98a | 99.91 |
| | | N | 185.92 ± 8.21a | 187.41 ± 8.85a | 3.32 ± 0.13a | 7.38 ± 1.92a | 99.9 |
| | | NP | 173.17 ± 8.04a | 171.69 ± 6.77a | 3.37 ± 0.05a | 6.34 ± 0.74a | 99.92 |
| | | M | 198.31 ± 6.25a | 196.54 ± 1.24a | 3.24 ± 0.31a | 9.05 ± 4.44a | 99.89 |
| 10–20 cm | Bacterial | CK | 2,547.23 ± 47.37a | 2,566.96 ± 65.75a | 6.21 ± 0.03a | 0.65 ± 0.06a | 97.09 |
| | | P | 2,495.59 ± 88.77a | 2,505.43 ± 104.85a | 6.22 ± 0.06a | 0.73 ± 0.10a | 97.22 |
| | | N | 2,508.50 ± 57.66a | 2,515.83 ± 35.21a | 6.16 ± 0.11a | 0.71 ± 0.15a | 97.08 |
| | | NP | 2,644.51 ± 2.29a | 2,649.42 ± 28.10a | 6.29 ± 0.05a | 0.57 ± 0.05a | 96.93 |
| | | M | 2,644.29 ± 64.92a | 2,640.09 ± 78.60a | 6.30 ± 0.09a | 0.56 ± 0.08a | 96.97 |
| | Fungi | CK | 158.80 ± 10.50a | 161.38 ± 9.69a | 3.22 ± 0.04a | 7.13 ± 0.41a | 99.92 |
| | | P | 149.36 ± 5.74a | 150.86 ± 7.12a | 3.38 ± 0.06a | 6.24 ± 0.54 | 99.95 |
| | | N | 167.98 ± 7.79a | 167.94 ± 7.30a | 3.33 ± 0.09a | 6.58 ± 1.05a | 99.93 |
| | | NP | 167.16 ± 9.63a | 165.80 ± 9.62a | 3.29 ± 0.04a | 7.14 ± 0.21a | 99.93 |
| | | M | 178.72 ± 11.88a | 176.12 ± 9.69a | 3.20 ± 0.12a | 8.16 ± 1.58a | 99.91 |

**Notes.**
Different lowercase letter after the data in each soil layer represented significance at $P < 0.05$ level (l.s.d.).
CK, Control; P, 60 kg P ha$^{-1}$; N, 100 kg N ha$^{-1}$; NP, 60 kg P ha$^{-1}$ plus 100 kg N ha$^{-1}$; M, 4,000 kg sheep manure ha$^{-1}$.

groups, with MF693140 clustered with *paraglomus occultum* in one group, and the other six sequences in another group. MF693141 was clustered with Ambispora fennica; MF693144 was clustered with Scutellospora species; MF693143 and MF693138 were clustered with Rhizophagus species; MF693139 was clustered with Diversispora species; MF693142 was clustered with Claroideglomus species. The abundance of MF693138 was the highest (Fig. 5). P fertilization significantly increased the abundance of MF693138, whereas N, NP or manure fertilization showed no significant difference when compared with the non-fertilized plot at both soil layers. The abundance of other AMF sequences was low and showed no variation between treatments.

AMF colonization was measured in three plant species, *C. squarrosa*, *S. krylovii* and *L. chinensis* (Table 4). The colonization (F%), colonization intensity (M%) and arbuscular (A%) varied among sampling months and plant species. Fertilization significantly influenced AMF colonization, however, no similar trend could be observed in different sampling months and plant species.

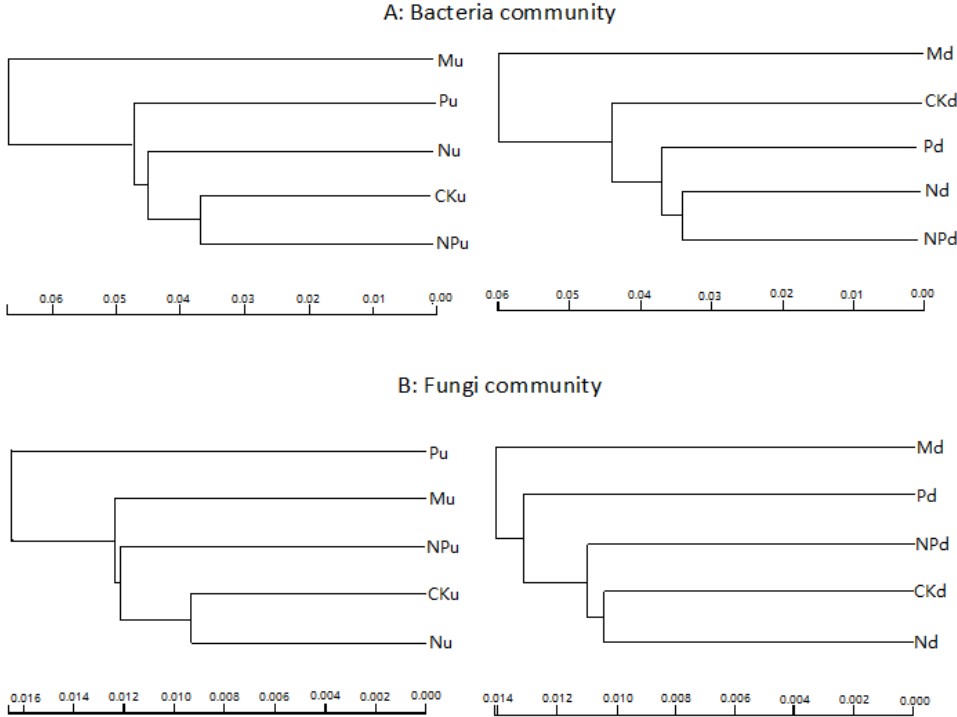

**Figure 2  Overall structural changes of bacterial communities (A) and fungi communities (B) from sequencing data as influenced by fertilizations.** The clustering was based on the weighted distance tree in Unifrac. CK, Control; P, 60 kg P ha$^{-1}$; N, 100 kg N ha$^{-1}$; NP, 60 kg P ha$^{-1}$ plus 100 kg N ha$^{-1}$; M, 4,000 kg sheep manure ha$^{-1}$. Lowercase letter "u" after treatment represented samples from 0–10 cm soil layer; "d" represented samples from 10–20 cm soil layer.

## Relationship between soil chemical properties and microbial community

Correlation Heatmap indicated that soil chemical properties significantly influenced the abundance of bacteria and fungi (Fig. 6). At 0–10 cm soil layer, cluster analysis separated soil pH with other environmental factors affecting the microbe abundance. The soil pH was positively correlated with the abundance of three phyla and negatively correlated with one phylum. The ratio of MBC/MBN was positively correlated with the abundance of two phyla and negatively correlated with two phyla. Both MBC and soil C/N were negatively correlated with three phyla. AN and MBN were further separated with other environmental factors. At 10–20 cm soil layer, MBN influenced more on the bacterial abundance, which was positively correlated with three phyla and negatively correlated with three phyla. C/N was positively correlated with two phyla and negatively correlated with three phyla. MBC/MBN, AN, AP and pH, were all correlated with one phylum, positively or negatively.

At 0–10 cm soil layer, factors affecting the abundance of fungi were separated into two groups, pH and other factors. Most of the soil chemical properties negatively influenced the fungi abundance, with the positive correlation was only observed with MBN (two phyla), C/N (one phylum), and MBC (one phylum). There were five and four fungi phyla

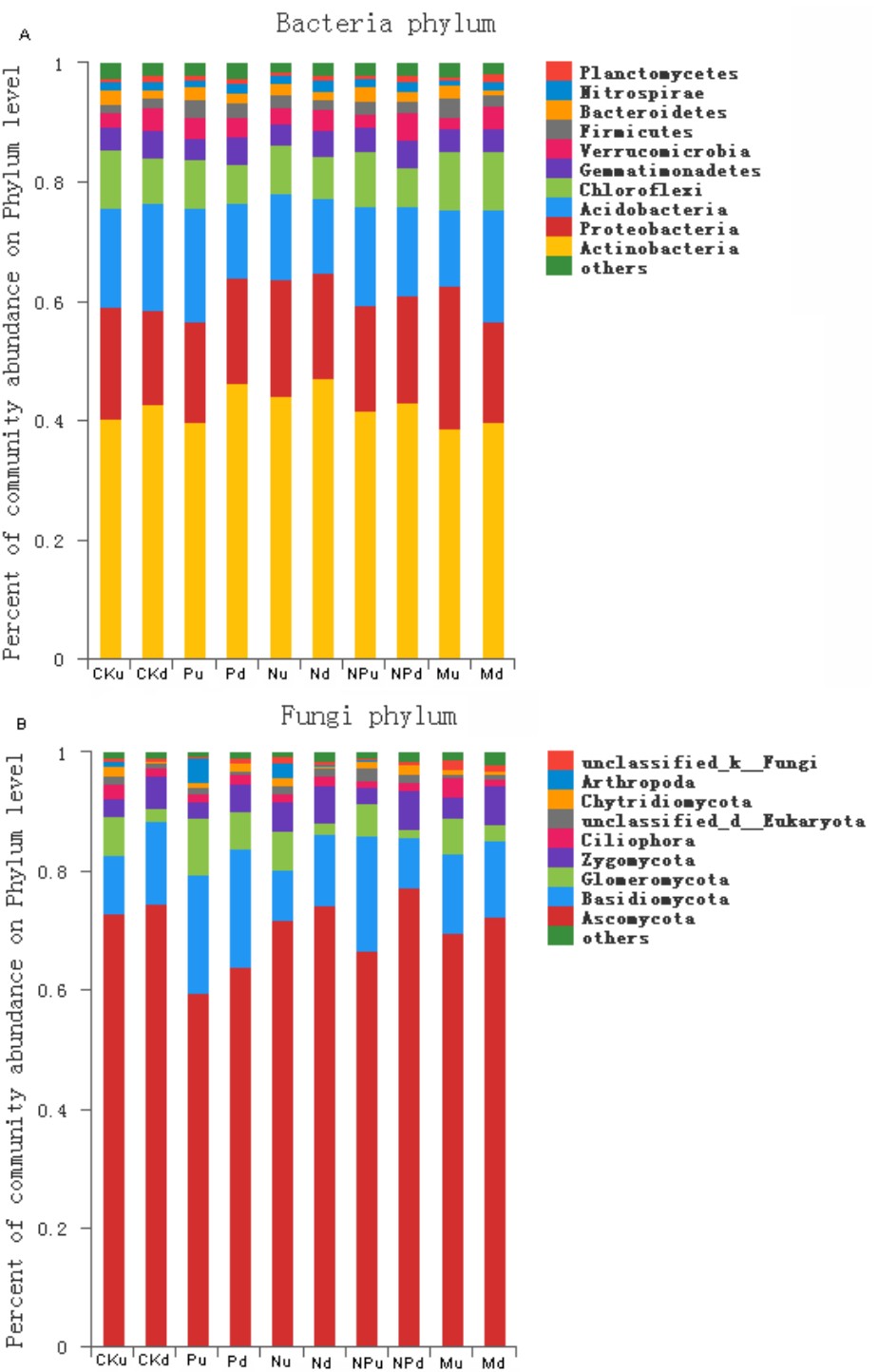

**Figure 3** **Proportions of the bacterial phyla (A) and fungal phyla (B) with the relative abundance higher than 1%.** CK, Control; P, 60 kg P ha$^{-1}$; N, 100 kg N ha$^{-1}$; NP, 60 kg P ha$^{-1}$ plus 100 kg N ha$^{-1}$; M, 4,000 kg sheep manure ha$^{-1}$. Lowercase letter "u" after treatment represented samples from 0–10 cm soil layer; "d" represented samples from 10–20 cm soil layer.

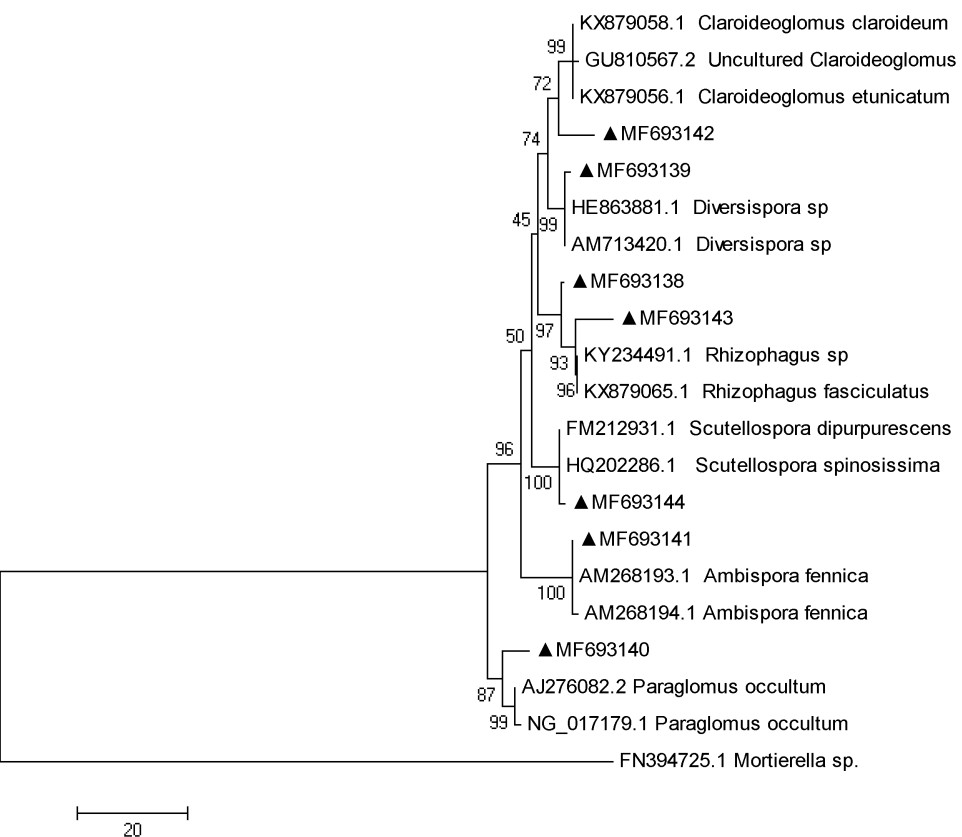

**Figure 4** **Phylogenetic tree inferred from the fungi sequences of AMF.** Numbers above branches indicate the bootstrap support (1,000 replicates). Cluster was conducted according to the maximum likelihood method. GenBank accession numbers followed by a black triangle (▲) represent the sequences obtained in the present study. Names followed by codes represent the sequences downloaded from GenBank.

were negatively correlated with AP and MBC/MBN, respectively. AN had no significant influence on any fungi phylum. At 10–20 cm, factors affecting the abundance of fungi were separated into two groups, with AN, MBN, MBC and MBC/MBN in one group and the others in second group. Factors in the first group were all positively correlated with fungi abundance excepting for a negative correlation with AN. Factors in the second group were mainly negatively correlated with the abundance of fungi excepting for a positive correlation with AP and pH.

## DISCUSSION

Fertilizations have been applied in many ecosystems to improve crop production or soil qualities (*Mitran et al., 2016*; *Paredes et al., 2011*). In the tested arid steppes, though the contents of soil AP increased significantly after P, NP or manure fertilizations after three years consecutive fertilizations, the aboveground biomass of the tested steppes was not improved, suggesting that sole fertilization in such ecosystem is not efficient in improving steppe productivity. This, on one hand, might be attributed to the low rainfall

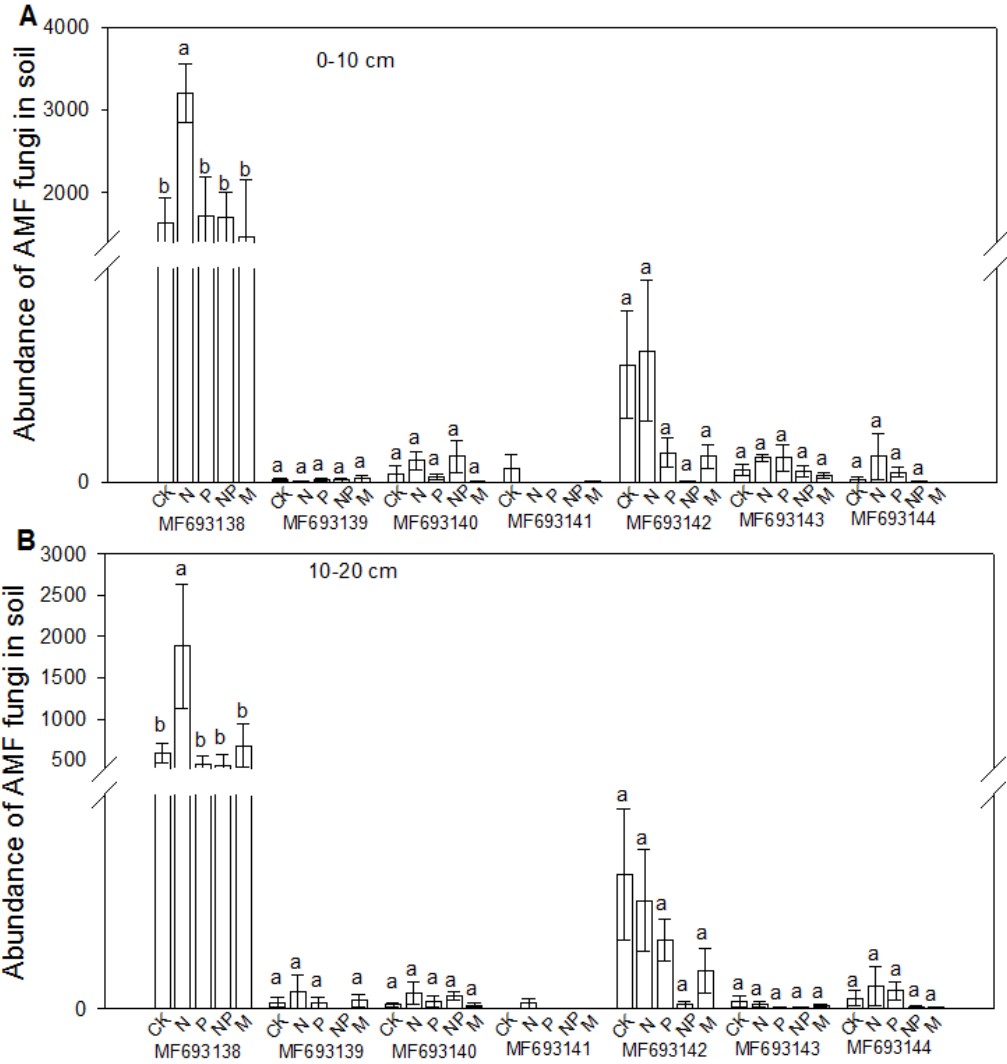

**Figure 5** **Effects of fertilizations on the abundance of AMF in soils at 0–10 cm (A) and 10–20 cm (B).**
CK, Control; P, 60 kg P ha$^{-1}$; N, 100 kg N ha$^{-1}$; NP, 60 kg P ha$^{-1}$ plus 100 kg N ha$^{-1}$; M, 4,000 kg sheep
manure ha$^{-1}$. Different lowercase letters above the data bar represented significance at $P < 0.05$ level
(l.s.d.).

during growing season (142 mm from May to July), which limited the fertilizer efficiency
(*Bai et al., 2007*; *Fang et al., 2005*), and thus the plant growth (*Austin, 2002*; *Jobbagy, Sala
& Paruelo, 2002*). In a degraded steppe in northern China, *Chen, Hooper & Lin (2011)*
also reported that nitrogen fertilization might only be effective in increasing rangeland
production in wet years. On the other hand, fertilization might alter other properties of
this ecosystem, such as soil microbe activities, which thus offset the positive effect of the
fertilizations (*Kautz, Wirth & Ellmer, 2004*; *Plaza et al., 2004*).

During the long-term evolution, the soil microbe and the plant species have adapted
to their growing environments, and their adaptation is relatively stable under certain
**Table 4** Effects of fertilization on AMF colonization of *Cleistogenes squarrosa, Stipa krylovii* and *Leymus chinensis* in different growing months.

| Month | Treatment | F% | | | M% | | | A% | | |
|---|---|---|---|---|---|---|---|---|---|---|
| | | *Cleistogenes squarrosa* | *Stipa krylovii* | *Leymus chinensis* | *Cleistogenes squarrosa* | *Stipa krylovii* | *Leymus chinensis* | *Cleistogenes squarrosa* | *Stipa krylovii* | *Leymus chinensis* |
| June | CK | 82.67 ± 2.40b | 92.00 ± 3.06a | 90.00 ± 2.31b | 7.59 ± 1.55b | 13.86 ± 5.30a | 12.62 ± 5.20b | 0.52 ± 0.08c | 2.16 ± 0.16b | 0.66 ± 0.36c |
| | P | 82.67 ± 2.40b | 91.33 ± 3.53a | 93.33 ± 3.53ab | 11.33 ± 2.26ab | 13.01 ± 2.05a | 13.37 ± 1.65b | 2.79 ± 0.37bc | 3.56 ± 1.15ab | 3.62 ± 0.58bc |
| | N | 88.67 ± 2.67ab | 95.33 ± 1.33a | 100.00 ± 0a | 17.03 ± 0.28a | 21.73 ± 4.42a | 20.00 ± 4.09ab | 5.41 ± 1.09a | 5.45 ± 1.62a | 4.70 ± 1.56b |
| | NP | 92.00 ± 1.15a | 90.00 ± 5.03a | 98.00 ± 1.15a | 16.34 ± 4.99ab | 8.15 ± 2.01a | 18.25 ± 2.21ab | 3.73 ± 1.14ab | 0.61 ± 0.06b | 5.25 ± 0.56b |
| | M | 83.33 ± 0.67b | 93.33 ± 1.33a | 94.67 ± 1.76ab | 10.93 ± 2.30ab | 13.21 ± 4.84a | 29.23 ± 4.82a | 0.78 ± 0.08c | 1.21 ± 0.32b | 11.33 ± 1.89a |
| July | CK | 90.67 ± 0.67c | 87.33 ± 0.67c | 100.00 ± 0a | 18.65 ± 1.20b | 7.13 ± 0.50c | 13.09 ± 0.05d | 14.06 ± 0.56b | 0.87 ± 0.05c | 5.54 ± 0.16d |
| | P | 96.67 ± 0.67b | 100.00 ± 0a | 100.00 ± 0a | 26.41 ± 0.54a | 43.25 ± 0.95b | 22.11 ± 1.26c | 18.14 ± 1.30a | 23.05 ± 0.73ab | 14.23 ± 0.18b |
| | N | 100.00 ± 0a | 98.67 ± 1.33a | 98.00 ± 1.15a | 17.16 ± 2.13b | 34.79 ± 8.91b | 34.75 ± 1.02a | 5.67 ± 0.10c | 17.36 ± 3.60b | 19.82 ± 0.14a |
| | NP | 82.67 ± 0.67e | 92.00 ± 1.15b | 90.67 ± 0.67b | 8.47 ± 0.30c | 13.11 ± 0.50c | 13.91 ± 0.14d | 3.06 ± 0.11d | 2.04 ± 0.25c | 7.00 ± 0.12c |
| | M | 88.67 ± 0.67d | 100.00 ± 0a | 98.67 ± 1.33a | 3.43 ± 0.65d | 58.17 ± 0.44a | 29.80 ± 1.04b | 0.88 ± 0.20e | 27.40 ± 3.25a | 14.53 ± 0.19b |
| August | CK | 91.67 ± 0.88b | 87.33 ± 0.67d | 98.67 ± 1.33a | 22.63 ± 2.17a | 20.95 ± 3.37b | 26.73 ± 1.22b | 2.94 ± 0.58b | 3.68 ± 1.50c | 14.51 ± 0.25a |
| | P | 96.67 ± 0.67a | 95.33 ± 0.67b | 98.00 ± 2.00a | 22.47 ± 0.94a | 36.41 ± 4.42a | 30.37 ± 0.33a | 16.27 ± 0.89a | 18.41 ± 1.63b | 12.00 ± 0.45b |
| | N | 88.67 ± 0.67c | 95.33 ± 0.67b | 98.67 ± 1.33a | 9.91 ± 0.54b | 25.68 ± 0.55b | 8.43 ± 1.26d | 2.91 ± 0.12b | 3.17 ± 0.55c | 6.41 ± 0.27d |
| | NP | 80.67 ± 0.67d | 91.33 ± 0.67c | 97.33 ± 1.33a | 7.67 ± 0.59b | 23.31 ± 0.34b | 19.29 ± 0.13c | 1.22 ± 0.09c | 4.38 ± 0.55c | 10.82 ± 0.07bc |
| | M | 91.33 ± 0.67b | 100.00 ± 0a | 96.00 ± 2.31a | 24.97 ± 0.52a | 40.53 ± 0.63a | 22.14 ± 0.95c | 3.73 ± 0.27b | 24.76 ± 0.93a | 9.97 ± 0.76c |

**Notes.**

F%, colonization; M%, colonization intensity; A%, arbuscular. Different lowercase letter after the data in each soil layer represented significance at $P < 0.05$ level (l.s.d). CK, Control; P, 60 kg P ha$^{-1}$; N, 100 kg N ha$^{-1}$; NP, 60 kg P ha$^{-1}$ plus 100 kg N ha$^{-1}$; M, 4,000 kg sheep manure ha$^{-1}$.

conditions (*Lau & Lennon, 2011*). In differently managed Belgian grasslands, *Denef et al. (2009)* reported that soil habitats significantly differed in microbial community structure as well as in gram-positive bacterial rhizodeposit-C uptake. In the current study, phosphorus, nitrogen and manure fertilizations increased MBC and MBN, suggesting that the propagation of soil microbes were improved when exterior nutrients were input into the degraded steppes. In a temperate desert, *Huang et al. (2018)* also reported that nitrogen fertilization increased microbial biomass by 14.0% and fungal biomass by 30.0% in winter. In central high land soils of Kenya, *Kamaa et al. (2011)* reported that organic inputs had a positive effect on both bacterial and fungal diversity with or without chemical fertilizers.

In total 3,927 OTU for bacteria and 453 OTU for fungi were identified from the tested soils. Fertilizations had no significant influence on the diversity and richness of soil bacteria or fungi at both soil layers. However, *Wang et al. (2017)* reported that inorganic fertilizer significantly decreased the Chao and ACE richness indexes of bacterial community, but increased that of the fungal community. *Farmer et al. (2017)* reported that long-term fertilizer addition greatly reduced the population of bacteria as well as its richness in a brown soil. The inconsistency between our results with these studies might be as a result of the differences in precipitation, soil type or fertilizer types or rates applied to the soil. In the Gurbantunggut Desert, Northwestern China, water addition increased bacterial and fungal diversity and abundance in summer but not in winter and spring (*Huang et al., 2018*). *Francioli et al. (2016)* reported that organic fertilization increased bacterial diversity, and stimulated microbial groups that were known to prefer nutrient-rich environments. These

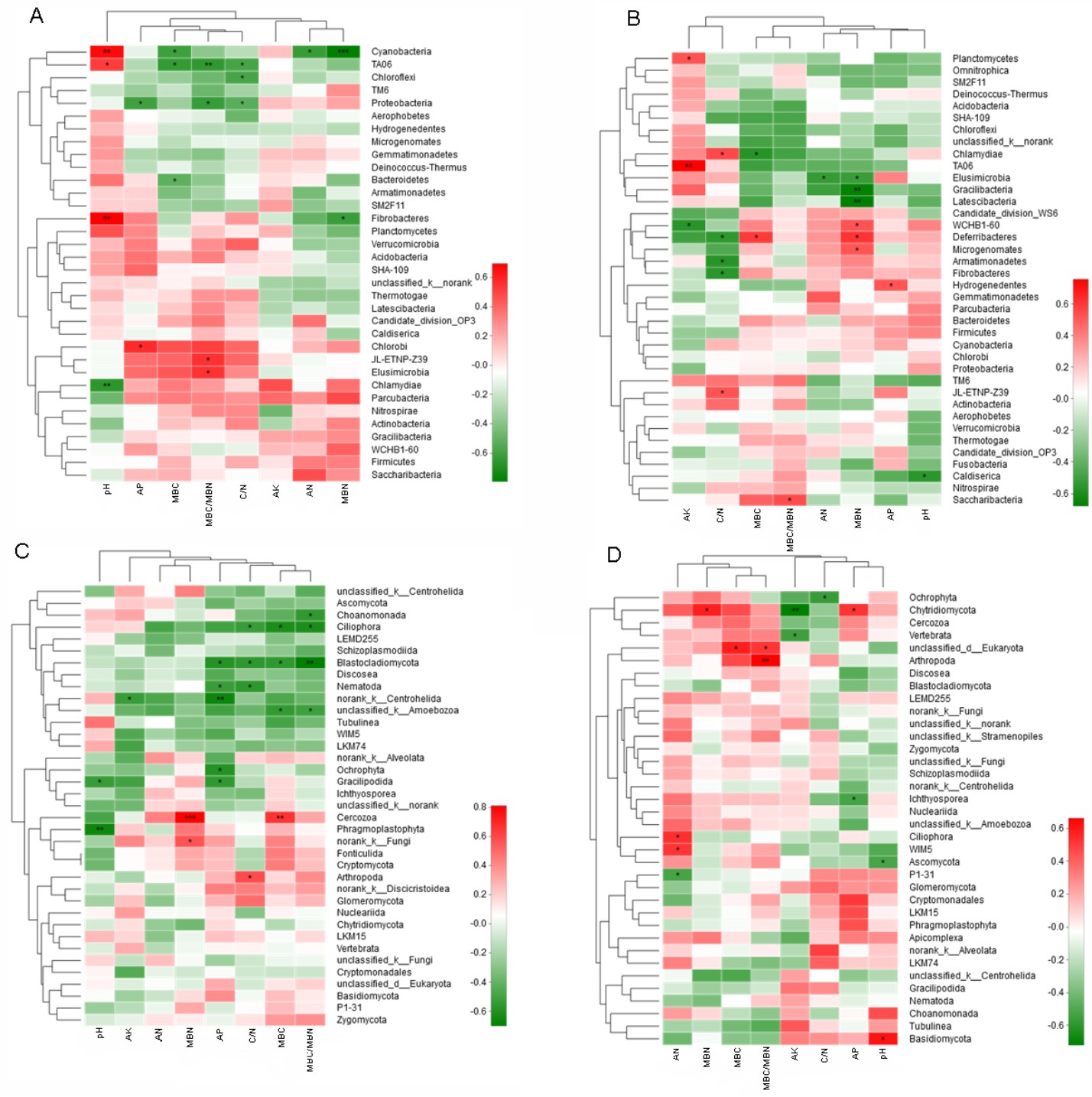

**Figure 6  Spearman Correlation Heatmap between bacterial and soil properties at 0–10 cm (A) and 10–20 cm (B) and fungi and soil properties at 0–10 cm (C) and 10–20 cm (D).** AN, alkaline hydrolyzed N; AP, available P; AK, available potassium; MBC, microbial biomass C; MBN, microbial biomass N. The *R* values are displayed in different colors and the right color card of the heat map is a color partition of different *R* values. $0.01 < P \leq 0.05^*$ $0.001 < P \leq 0.01^{**}$ $P \leq 0.001^{***}$. CK, Control; P, 60 kg P ha$^{-1}$; N, 100 kg N ha$^{-1}$; NP, 60 kg P ha$^{-1}$ plus 100 kg N ha$^{-1}$; M, 4,000 kg sheep manure ha$^{-1}$. Lowercase letter "u" after treatment represented samples from 0–10 cm soil layer; "d" represented samples from 10–20 cm soil layer.

results suggested that the responses of the abundance of bacteria and fungi to fertilizations might be differed among microbe species, causing insignificant changes of the total richness and diversity in arid steppes.

Cluster analysis based on the sequence data indicated that bacterial and fungi communities in manure fertilized plots were separated with other fertilizations at both soil layers, excepting fungi communities in P fertilized plots which were separated with other fertilizations at 0–10 cm soil layer. This suggested that the sensitivities of the steppe soil microbes differed in their responses to soil nutrients and fertilizer types. *Wang, Ji & Gao (2016)* reported that the prokaryotic community structure in manure fertilized soil was different from that in N and P and NP fertilized soils. Long-term application of inorganic fertilizer resulted in a decrease of bacterial diversity, whereas manure application increased microbe diversity (*Sapp et al., 2015*; *Sun et al., 2015*).

In the present study, high-throughput sequencing revealed that the predominant bacteria phyla were Acidobacteria, Proteobacteria, Actinobacteria and Chloroflexi, accounting for an average 84.63% of the total bacterial sequences; the predominant fungal phylum was Ascomycota, which accounted for an average 71.26% of the total fungal sequences. These results were consistent with the previous studies based on agricultural soils (*Feng et al., 2015*; *Su et al., 2017*; *Wu et al., 2011b*). Further analysis confirmed that the abundance of soil bacteria and fungi phylum differed in their responses to fertilizers. For example, nitrogen fertilization increased the abundance of Nitrospira and gram negative bacteria, among which Nitrifier oxidizes the nitrite into nitrate, benefiting the plant nitrogen use efficiency (*Kant, 2017*). Manure relatively increased the abundance of Proteobacteria, which might be attributed to the higher organic carbon input form manures. *Fierer, Bradford & Jackson (2007)* reported that Proteobacteria, especially the Betaproteobacteria were attributed and favored by nutrient-rich conditions with high C content. At 10–20 cm soil layer, manure fertilizations relatively increased the abundance of Chloroflexi, facultative anaerobic bacteria, among which Chlorobi has also been shown to respond differently to the N and P inputs in a steppe ecosystem (*Pan et al., 2014*). For fungi, manure increased the abundance of Ciliophora, P and NP increased the abundance of Chytridiomycota, and P increased the abundance of AMF MF693138. These fungi might differ in their nutrient requirements during propagation (*Dong & Yao, 2005*; *Zhu et al., 2008*), and the variations of soil chemical properties caused after fertilization resulted in such response differences.

AMF are widely distributed in natural soils, which improve plant nutrient uptake and drought resistances (*Valyi, Rillig & Hempel, 2015*; *Tilman, Wedin & Knops, 1996*). In the tested soils, in total seven AMF sequences were identified, being clustered with *Paraglomus occultum*, Ambispora fennica, Scutellospora species, Rhizophagus species, Diversispora, or Claroideglomus species. Though most of these AMF species were also observed in other steppes (*Wang et al., 2014*; *Hiiesalu et al., 2014*), no Glomus species was identified in the tested soils, *G. mosseae* of which has been reported from all continents except the Antarctic (*Guo et al., 2016*; *Rosendahl, McGee & Morton, 2009*), reflecting its wide adaptation and positive roles in different ecosystems (*Smith & Read, 1997*). The possible reason might be related to the soil properties of the current degraded steppe. A study with non-degraded, moderately degraded and severely degraded steppes in Inner Mongolia has shown that

different AMF species showed different distributions among steppes, suggesting that both biotic and abiotic factors were important in determining the AMF communities (*Tian et al., 2009*). Fertilization, particularly the P fertilizer, influenced the abundance of the AMF species in the tested steppes, confirming that the AMF diversity was driven by soil properties. Variations of AMF colonization were also observed between sampling months and fertilizations. However, no similar trend could be observed in different months and plant species. This suggested that the AMF species colonizing each plant might be different between plant species (*Guo et al., 2016*), which attributed to their response variations to fertilizations.

In many ecological systems, soil pH is a very important ecological factor for soil microbes (*Fierer & Jackson, 2006*; *Rousk, Brookes & Baath, 2010*). Heatmap analysis further indicated that soil pH in the tested steppe was also one of the most important soil properties influencing the abundance of soil bacteria and fungi. Though fertilization had no significant influence on soil pH, soil pH differed between two soil layers, causing the difference of microbial communities. Besides soil pH, other soil chemical properties, such as the concentrations of AP, AN, AK, C/N, MBN, MBC, and MBC/MBN all had positive or negative influences on bacterial and fungi communities. *Farmer et al. (2017)* reported that soil phosphorus and pH positively influenced the bacterial richness. In a semi-arid steppe, N inputs significantly increased the relative abundance of Proteobacteria and Firmicutes but reduced the abundance of Acidobacteria, Nitrospirae and Chloraflexi, whereas P additions significantly affected Armatimonadetes and Chlorobi (*Ling et al., 2017*). These results suggested that soil bacteria and fungi communities in degraded steppes could be altered by improving the soil chemical properties through fertilizations. During nature restoration the efficiency of nutrient cycling can increase by shift in microorganism composition (*Morrien et al., 2017*). Three years' fertilization improved soil qualities and altered soil bacteria and fungi communities; however, it is still a long term procedure to restore such degraded steppe into original status, including soil properties (*McLauchlan, Hobbie & Post, 2006*), plant species composition (*Zhan, Li & Cheng, 2007*), and the network between soil microbe and plants (*Hamonts et al., 2017*).

## CONCLUSIONS

Fertilizations had no significant influence on the richness and diversity of the bacteria and fungi. However, the abundance of individual bacterial or fungi species was sensitive to fertilizations, particularly to manure application, mainly attributing to the variations of soil chemical properties caused by fertilizations. Fertilization, particularly the P fertilization, influenced the abundance of the AMF species and the AMF colonization in the tested steppes, suggesting that the AMF diversity was driven by soil properties. Among the soil properties, soil pH in the tested steppe was one of the most important soil properties influencing the abundance of soil bacteria and fungi. Soil bacteria and fungi communities in degraded steppes could be altered by improving the soil chemical properties through fertilizations. However, it is still not clear whether the alteration of the microbe community is detrimental or beneficial to the degraded arid steppes.

**Abbreviations**

| | |
|---|---|
| **AMF** | arbuscular mycorrhizal fungi |
| **AN** | alkali dispelled nitrogen |
| **AOA** | ammonia-oxidizing archaea |
| **AOB** | ammonia-oxidizing bacteria |
| **AP** | available phosphorus |
| **MBC** | microbial biomass carbon |
| **MBN** | microbial biomass nitrogen |
| **OTUs** | operational taxonomic units |
| **TN** | total nitrogen |
| **OC** | organic carbon |
| **TP** | total phosphorus |

## ACKNOWLEDGEMENTS
We appreciate the help from Mr. Jianjun Chen for field sampling and fertilization.

### Funding
The work was supported by the National Key Basic Research Program of China (No. 2014CB138806) and National Natural Science Foundation of China (No. 31670407). The funders had no role in study design, data collection and analysis, decision to publish, or preparation of the manuscript.

### Grant Disclosures
The following grant information was disclosed by the authors:
National Key Basic Research Program of China: 2014CB138806.
National Natural Science Foundation of China: 31670407.

### Competing Interests
The authors declare there are no competing interests.

### Author Contributions

- Luhua Yao performed the experiments, analyzed the data, contributed reagents/materials/analysis tools, prepared figures and/or tables, authored or reviewed drafts of the paper, approved the final draft.
- Dangjun Wang and Lin Kang performed the experiments, analyzed the data, authored or reviewed drafts of the paper, approved the final draft.
- Dengke Wang performed the experiments, authored or reviewed drafts of the paper, approved the final draft.
- Yong Zhang conceived and designed the experiments, performed the experiments, contributed reagents/materials/analysis tools, authored or reviewed drafts of the paper, approved the final draft.

- Xiangyang Hou conceived and designed the experiments, authored or reviewed drafts of the paper, approved the final draft.
- Yanjun Guo conceived and designed the experiments, analyzed the data, contributed reagents/materials/analysis tools, prepared figures and/or tables, authored or reviewed drafts of the paper, approved the final draft.

## Data Availability

   NCBI Sequence Read Archive database, SRP126759.

## Supplemental Information

Supplemental information for this article can be found online at http://dx.doi.org/10.7717/peerj.4623#supplemental-information.

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
