# Peer review of "Effects of fertilizations on soil bacteria and fungi communities in a degraded arid steppe revealed by high through-put sequencing"

_PeerJ, doi:10.7717/peerj.4623_

## Round 0.1 · original submission · Major Revisions

Please revise the manuscript carefully taking all comments of the reviewers into consideration and then resubmit it for a second review.

Reviewer 1 ·

Basic reporting

Well, This study is mainly focused on the effects of responses of above ground biomass and soil chemical and biological properties. Then data from different sampling month is not necessary. Only focused on the samples which can offer the information of your aim. I strongly suggest to restructure this manuscript and do the related discussion afterwards. Conclusion is fine but need to be edited if “Results and Discussion” changed. Tables and Figures (resolution) need to be checked based on the requirement of PeerJ. Constant format of Unit should be given in whole manuscript.

Experimental design

Experiment design and sampling strategies seems strange not well-expressed. Line 161: why sampled in June July and August, 2016? and Line 165: why only samples from July did the DNA extraction?

Validity of the findings

It is the interesting study. However, language should be corrected. Results and discussion can be focused on your objective.

Additional comments

It is the interesting study. However, language should be corrected. Constant format of Unit should be given in the whole manuscript. Experiment design and sampling strategies seems strange and not well-expressed. Tables and Figures (resolution) need to be checked based on the requirement of PeerJ. Some detail comments will be added as below.
Line 23-30: It is not really methods but only all the tested parameters. Rephrase. Meanwhile, you mentioned fertilization, but which specific fertilization you focused on should also be added.
Line 29:add space between “and” and “5”.
Line 31: OUT? The explanation in the blanket is not the same as you described in the note part of previous page. And the format of blanket is not correct.
Line 37-38: soil chemical properties? which are they?
Line 43-45: “It was suggested that no fertilization was recommended to restore such degraded arid steppe in dry years.” Very strange conclusion. You didn’t refer any information about dry or wet year. I would say your findings can not support this conclusion (I assume it is your conclusion otherwise you need to add it), especially for such “dry year”.
Line 48: “Degradation” is listed as keyword but it is not specific. Can be changed to “Soil degradation”.
Line 75: check the citation. terHorst ?? Ter Horst?
Line 105: Well, you referred “Metagenomic approaches” but no clear information or review you did about this approach contributes to your study. That means some literature review need to be done in this point.
Line 110-115. These information can be moved to “Methods” section.
Line 114: MBN and MBC should be explained here as they present in the first time.
Line 161: why sampled in June July and August, 2016?
Line 165: why only samples from July did the DNA extraction?
Line 167-187: Only TOC, AP AN, pH, MBC and MBN were tested. The title can be changed to “Tested soil properties” instead of “Soil chemical analysis”.
Line 182-187: If you want to show the equation of MBC and MBN, please use the mathematic way to describe. All the symbols you mention in the equation should be defined and united.
Line 217-236: can be moved to “Statistical analysis” section.
Line 237: “Analysis” into “analysis”
Results and discussion can be focused on your objective. Well, it is mainly focused on the effects of responses of above ground biomass and soil chemical and biological properties. Then data from different sampling month is not necessary. Only focused on the samples which can offer the information of your aim. I strongly suggest to restructure this manuscript and do the related discussion afterwards. Conclusion is fine but need to be edited if “Results and Discussion” changed.

Reviewer 2 ·

Basic reporting

I highly recommend to review the English writing by a native English speaking colleague because often it is difficult to follow the ideas the authors try to communicate. For example: text in lines 78-80, 325,
The literature references presented concerning use of inorganic fertilizers to restore degraded land are mainly regional. I suggest the authors to search for references on the topic from somewhere else.
Figures can be improved: In general: the Title should be one sentence title of a figure (instructions for authors) and the Legend is the optional several sentence description. In particular: There are imprecisions in the description of the Results, in line 275 it is written that for MBC, N fertilized plot was significantly higher than other treatments at 10-20 cm layer, it is not N but P. In the legend and on the figure itself (Fig 2), is not needed to write u and d for 0-10 cm and 10-20 cm soil layers respectively because on the figure it is indicated the 0-10 and 10-20 cm layers. However, the authors should check their data because the text for Fig. 2, do not correspond with what it is shown on the figure, see commented (marked) lines 298-300.
The legend of Fig 4. is incorrect, it belongs to Fig 6. and vice versa, the legend of Fig 6. belongs to Fig 4.
Likewise Tables need to be corrected: In the legend of Table 1, it is not explicit the comparison among months and soil layers, but in the text it is written about differences and not differences among months and layers: line 252.
Tables and Figures need to be deeply revised because there are so many imprecisions in the text.

Experimental design

The main scientific contribution of the present paper would be the effect of fertilizer application on diversity and abundance of soil bacteria and fungi and colonization of arbuscular mycorrhizal fungi (AMF) on a degraded ecosystem. However, the authors did not include in their experiment, reference plots of non-degraded steppe. Restoration of a degraded land aims to convert the land to the original unperturbed ecosystem. It has been found that after perturbation and into natural succession soil microorganisms change towards its original status. It implies that during nature restoration the efficiency of nutrient cycling can increase by shift in microorganism composition. I suggest to the authors to read the following article: Morriën et. al. 2017. Soil networks become more connected and take up more carbon as nature restoration progresses. Nature communications 8:14349 | DOI: 10.1038/ncomms14349 |www.nature.com/naturecommunications.
The use of inorganic fertilizers is not common in the restoration of degraded ecosystem. Although, the authors present some references about the negative effects of inorganic fertilizers to nature, their results in this topic would be their main contribution to knowledge. However, there were not differences among treatments. The authors acknowledge that three years of trial is probably too short time to identify differences among treatments.
There are some confusions and enquiries to solve concerning the methods used in the present experiment: under the Soil chemical analysis section, the authors write the methods for organic C, total N, alkali dispersed N, available P and pH. However, there is not mention to the methods used for the analyses of the soil samples taken before the start of the trial. Except, for available P (line 138); “analyzed by the Olsen method”. However, later, the authors describe (lines 171-173) under Soil chemical analysis the Olsen-P method without mention it. On line 168 the authors write Total organic carbon, it is not needed to write Total. Total carbon is the addition of organic carbon and inorganic carbon. Lines 170-171 it is written Kjeldahl method without explaining that this method is to determine total nitrogen.

Validity of the findings

As I wrote above, the authors did not include in their experiment reference plots of non-degraded steppe. To validate such an experiment it is needed to include reference plots of non-degraded steppe with the same number of replicates as the rest of the treatments. In the same way, the authors recognize, that it is recommended longer experimental trials to validate or reject the effect of fertilizer application on above ground biomass and soil chemical and biological properties on degraded arid steppes.
For the data to be robust it is needed to include plots of non-degraded steppe and a longer experimental trial.
Conclusions are correct, “Three years consecutive fertilizations in degraded arid steppes had no significant influences on the most soil chemical properties and the aboveground biomass”. However, the experiment as it was designed does not permit to demonstrate that fertilization in degraded arid steppes have or not have influence on chemical properties, richness and diversity of the bacteria and fungi and the aboveground biomass.

Additional comments

I would suggest the authors to correct the manuscript for the English writing and peer through the entire paper for basic writing and structure errors, for example do not present tables and figures in the wrong order and label them with the wrong title and legend, as it was the case with figures 4 and 6. Likewise, check over the figures and tables in order to make explicit what it is written in the text of the manuscript, I made some remarks in the text concerning this.

---

## Round 0.2 · accepted · Accept

The manuscript has improved and can now be accepted.

Reviewer 1 ·

Basic reporting

The manuscript was well-revised. All the comments were responsd one by one. In my opinion, all revision and response are acceptable.

Experimental design

Well, in this revised manuscript, this part is improved a lot and would be clear to support your findings.

Validity of the findings

no comment

Additional comments

The revised manuscript was improved. With the revised title of this manuscript, authors did very good job. Now only some editing work should be done for publishing in PeerJ.